# Automated Age-Related Macular Degeneration Detector on Optical Coherence Tomography Images Using Slice-Sum Local Binary Patterns and Support Vector Machine

**DOI:** 10.3390/s23177315

**Published:** 2023-08-22

**Authors:** Yao-Wen Yu, Cheng-Hung Lin, Cheng-Kai Lu, Jia-Kang Wang, Tzu-Lun Huang

**Affiliations:** 1Department of Electrical Engineering, Yuan Ze University, Taoyuan City 320, Taiwan; 2Department of Electrical Engineering, National Taiwan Normal University, Taipei City 106, Taiwan; cklu@ntnu.edu.tw; 3Department of Ophthalmology, Far Eastern Memorial Hospital, New Taipei City 220, Taiwan

**Keywords:** age-related macular degeneration (AMD), artificial intelligence (AI), application-specific integrated circuit (ASIC), optical coherence tomography (OCT), slice-sum

## Abstract

Artificial intelligence has revolutionised smart medicine, resulting in enhanced medical care. This study presents an automated detector chip for age-related macular degeneration (AMD) using a support vector machine (SVM) and three-dimensional (3D) optical coherence tomography (OCT) volume. The aim is to assist ophthalmologists by reducing the time-consuming AMD medical examination. Using the property of 3D OCT volume, a modified feature vector connected method called slice-sum is proposed, reducing computational complexity while maintaining high detection accuracy. Compared to previous methods, this method significantly reduces computational complexity by at least a hundredfold. Image adjustment and noise removal steps are excluded for classification accuracy, and the feature extraction algorithm of local binary patterns is determined based on hardware consumption considerations. Through optimisation of the feature vector connection method after feature extraction, the computational complexity of SVM detection is significantly reduced, making it applicable to similar 3D datasets. Additionally, the design supports model replacement, allowing users to train and update classification models as needed. Using TSMC 40 nm CMOS technology, the proposed detector achieves a core area of 0.12 mm^2^ while demonstrating a classification throughput of 8.87 decisions/s at a maximum operating frequency of 454.54 MHz. The detector achieves a final testing classification accuracy of 92.31%.

## 1. Introduction

In recent times, the remarkable growth of artificial intelligence (AI) has led to advancements in smart healthcare applications across various medical domains [1]. The integration of AI and healthcare has played a crucial role in fostering partnerships for the goals of sustainable development. The smart medical applications have made significant contributions, enabling the identification of pathology differences that doctors may find challenging to diagnose [2,3,4,5,6,7,8].

Nowadays, ophthalmologists frequently use fundus colour and optical coherence tomography (OCT) images to diagnose macular diseases [8]. Fundus colour images provide two-dimensional information, including details of the topmost layer of the retina, retinal vasculature, inner limiting membrane, and other underlying layers. In comparison, OCT imaging is a non-invasive optical technique that offers excellent spatial resolution and tissue distribution clarity. In addition, OCT scanners are no longer confined to large-scale scanning instruments found in major hospitals. Portable OCT scanners [9,10] that can be used by medical staff for medical patrol are gradually being developed. While hyperspectral and multispectral imaging systems are not typically used as primary tools for disease diagnosis in the medical context, they can complement other imaging modalities, including OCT. For example, they may provide additional information about tissue composition or enhance the detection of specific features. The combination of different imaging techniques can potentially improve diagnostic accuracy and provide a more comprehensive understanding of tissue properties [11].

Automated age-related macular degeneration (AMD) is the most common macular disease affecting individuals over the age of fifty [12]. As individuals grow older, the vertebral cells, located at the centre of the retina, degenerate and lose their ability to absorb nutrients; this leads to the atrophy of the macula. Additionally, abnormal hyperplasia of choroidal blood vessels develops beneath the macula, making these newly formed vessels susceptible to rupture. The rupture of these vessels causes bleeding or exudation, leading to macular swelling, as shown in Figure 1. Over time, these factors contribute to a gradual decline in vision.

The vision is an important and sensitive organ in human beings, and even the slightest discomfort can prompt individuals to visit the hospital for an eye examination. However, diagnosing eye diseases presents significant challenges for ophthalmologists. In addition, rural areas often lack adequate medical resources and doctors, making it necessary for people to travel long distances to reach hospitals in cities. This imposes a heavy burden on remote villagers and can lead to significant medical delays [13]. Therefore, it is crucial to focus on developing an automated eye disease detection system using an edge-design chip with a portable OCT scanner [6]. By employing this approach, if the detection result is positive, the patient can then be referred to the hospital for a more comprehensive eye examination. Developing a complete and credible automated system for detecting eye diseases would benefit both doctors and patients, creating a win–win situation [7].

In this study, we propose an automated AMD detector to assist ophthalmologists by reducing excessive and tedious medical examinations. To support medical service in rural areas, it can be implemented on a single edge-design chip and integrated into a portable OCT scanner, with the aim of promoting good health and well-being. Compared with deep learning (DL) classifiers, low-complexity and hardware-friendly machine learning (ML) classifiers are suitable for automated AMD detection designed for a portable OCT scanner. Therefore, the proposed automated AMD detector utilizes a support vector machine (SVM) [14] classifier because the SVM is a well-established supervised machine learning (ML) algorithm known for its simplicity, quick classifier ability, and excellent performance. Additionally, the proposed detector employs local binary patterns (LBP) [15] to extract features because of its low computational complexity. The relatively complicated task of training the classifier model was performed on software to develop an edge chip design with low computational complexity. Subsequently, the pre-trained parameters and input of the unknown OCT data can be applied into an edge chip, obtaining the classifier result. In this study, the proposed detector has the following contributions:(a)The low-complexity feature vector connection method, called slice-sum, is proposed to reduce the computational processing required by the SVM classifier. Consequently, the method significantly reduces computational costs by at least a hundredfold compared to previous methods;(b)The detector uses only the LBP and SVM classifier, which minimises the hardware area cost for image processing. This feature is advantageous for integrating the detector into a portable OCT scanner;(c)The model parameters of the proposed detector have been designed to be replaceable, ensuring that users can easily update the model as per their future requirements.

To verify the proposed automated AMD detector, a prototype chip has been implemented, occupying a core area of 0.12 mm^2^, using the TSMC 40 nm CMOS technology. The detector achieved a classification accuracy of 92.31%. In terms of performance, the hardware operation time for predicting a single OCT volume was 139 times faster than the MATLAB software. Furthermore, the classification throughput reached 8.87 decisions per second, making it suitable for integration into portable OCT scanners available in the market. To the best of our knowledge, we present the first instance of single-chip implementation for automated AMD detection in the literature.

The remainder of this study is organised as follows: Section 2 provides a review of the automated eye disease detection systems. Section 3 presents the steps for improving the proposed system, with a description of the adopted algorithms in the following subsections. Section 4 presents the VLSI implementation of the proposed detector, including the chip results and a comprehensive work comparison. Finally, Section 5 provides a conclusion to this study.

## 2. Fundamentals of Automated Eye Disease Detection System

This section describes the related works of automated eye disease detection systems and reviews the fundamental flow, including image pre-processing, feature extraction, training, and classification.

### 2.1. Related Works

Recently, AI techniques have been extensively utilized to process human retinal information on OCT images. The tasks include DL-based denoising [16,17], DL-based segmentation [18,19], DL-based disease classification [6,7,8,20,21,22], and ML-based disease classification [3,6,23,24,25,26,27,28]. For the DL-based disease classification, the study in [6] has utilized the AlexNet, GoogLeNet, and Inception-ResNet-v2 to achieve high AMD detection accuracies up to 97.39%, 96.41%, and 98.18%, respectively. However, it also revealed that the DL-based disease classification consumes high computational complexity, using powerful graphic processing units (GPUs) of NVDIA GeForce GTX 1070 to achieve a high detection accuracy compared to the SVM-based disease classification using the Intel i7-8700 3.2 GHz 6-core processor only. It means that the current DL-based AMD detections executed by high-level GPUs are not a cost-effective hardware implementation. This initially motivated us to design a low-cost automated AMD detector using the SVM classifier since the SVM exhibits a lower area cost and becomes suitable for single-edge chip implementation [5,29]. In this study, the state-of-the-art SVM-based eye disease classifications are discussed as follows while the other AI tasks on OCT images can be referred to the aforementioned citations for details.

Table 1 lists the state-of-the-art studies related to SVM-based eye disease classifications. It is worth noting that the studies in [3,24,26,28] employed image-level classification while the studies in [25,27] employed volume-level classification. The volume-level classification is more practical than the image-level classification since an OCT volume is representative for a whole subject. In 2011, Liu et al. proposed a comprehensive automated eye disease framework and still achieved good performance utilizing LBP and SVM for classifying AMD, macular edema (ME), and macular hole (MH) [24]. The area under the receiver operator characteristic curve (AUC) was used as the classification index, which differs from the commonly adopted classification accuracy (ACC). To classify normal macula and diabetic macular edema (DME) OCT volumes, the study in [25] denoised the OCT images using a three-dimension filter (BM3DF) [30] and extracted feature vectors using the LBP combined with a histogram of oriented gradients (HOG) [31]. The classification capability is evaluated through sensitivity (SE) and specificity (SP). Meanwhile, the study in [26] demonstrated the effectiveness of transfer learning (TL) using GoogLeNet to classify normal macula, AMD, and DME OCT images, achieving a high image-level accuracy of 94.0%. Subsequently, employing distinct convolutional neural networks (CNNs) for TL to extract feature vectors from OCT images has been widely adopted for the SVM classifier [3,27,28]. Nonetheless, as noted in [6], the computational complexity of the LBP remains significantly lower than that of the TL involving CNNs, which are still executed by computation-intensive GPUs. Notably, the recent studies in [3,27,28] have not removed the speckle noise for the feature extraction and still attained commendable classification accuracies. Although the automated eye disease classifications listed in Table 1 have achieved high classification accuracies, they have only been implemented through software rather than a low-cost single chip, which is suitably integrated into a portable OCT scanner. After a review of the-state-of-art literature, the basic flow of the automated eye disease detection system depicted in Figure 2 is elaborated on in the subsequent subsections.

### 2.2. Image Pre-Processing

Image pre-processing involves two main methods: adjusting the image size and removing noise from images. There are several techniques for noise removal, including BM3DF, hybrid median filter (HMF) [32], and adaptive wiener filter (AWF) [33].

Regarding the previous topic, the classification accuracy is nearly the same whether the image-cropping step is used or not [23]. However, the analysis of computational time reduction focuses on the software phase rather than the hardware phase. Considering the hardware aspect, image cropping only increases the area cost without any improvement in classification accuracy.

In contrast, while BM3DF is well-known for its powerful image noise removal capabilities, resulting in excellent clarity for human looks, its target audience is not ophthalmologists but rather AI detection systems for automated eye detection. Owing to its high complexity, BM3DF does not offer an advantage in classification accuracy. High classification accuracy can still be achieved even without noise removal [6]. Moreover, the HMF exhibits a slight increase in accuracy while maintaining a friendly hardware complexity. A combined architectural method using the HMF and feature extraction was proposed [34].

### 2.3. Feature Extraction

The primary objective of feature extraction is to effectively differentiate the features present in AMD and normal OCT images. Prior studies have employed various approaches, such as LBP [24,25] and HOG [25]. Although these studies have different classification accuracies, it is not possible to directly compare them in terms of feature extraction owing to variations in experimental conditions, including image pre-processing, training flow, and dataset.

### 2.4. Training and Classification

A simple training flow for an SVM model using the MATLAB toolbox was described in [35]. This process involved *K*-Fold cross-validation, which helps prevent overfitting. The derivation of *K*-Fold cross-validation is as follows: first, all OCT volumes in the dataset were randomly separated into *K* subdatasets. The first subdataset, $F_{1}$, served as the testing data while $F_{2}$ to $F_{k}$ constituted the training data employed for building the classification model, as shown in Figure 3. Next, $F_{2}$ would serve as the testing data, with $F_{1}$ and $F_{3}$ to $F_{k}$ serving as the training data. This progression continued until all subdatasets had been used as testing data, resulting in *K* testing results. Finally, the average of these *K* test results was considered as the final *K*-fold accuracy of the model.

## 3. Proposed Low-Complexity Automated AMD Detector

This section presents the proposed automated AMD detection system and the training dataset. The simulation results are also demonstrated here before introducing the VLSI implementation of the automated AMD detector.

### 3.1. OCT Dataset

The OCT dataset we used was obtained from the open resources of Duke University [36]. This dataset registered at ClinicalTrials.gov with an identifier of NCT00734487 comprises 269 AMD and 115 normal OCT volumes. Each OCT volume consists of 100 OCT images, each with a size of 512 × 1000 pixels and greyscale pixel values, as shown in Figure 4.

### 3.2. Local Binary Patterns

LBP are a practical feature extraction algorithm commonly employed for grey-scale images. It effectively represents texture features through its histogram statistics method. The classic LBP algorithm is as follows:(1)$${LBP}_{p,r}=\sum_{i=1}^{p}u\left(g_{i}-g_{c}\right)2^{p-1},$$
with
(2)$$u\left(x\right)=\left\{\begin{array}{c} 1,\;x\geq 0\\ 0,x<0 \end{array}\right\},$$
where p denotes the number of image pixel sampling points, and r represents the radius of the window used for calculations. In this equation, $g_{i}$ refers to the pixel value of the neighbouring sampling point, while $g_{c}$ is the pixel value of the centre in the window being considered. The derivation of the LBP algorithm is illustrated in Figure 5, which depicts a comparison of pixel values between the neighbour points and the centre point. Consequently, a feature vector [00001111] is obtained, which is sorted counterclockwise and serves as a representation of the texture feature for the specific 3 × 3 window being analysed.

The histogram statistics subsequently represent the occurrence of features in the entire image. The number of feature types in the histogram depends on the sampling points. For example, if we have p=8 sampling points, which corresponds to $2^{8}$, we will have 256 feature types. Along with the translation of the calculated window in the whole image. This enables us to obtain a final feature vector that represents the texture feature of a single OCT image, as shown in Figure 6.

The introduction to LBP mentioned above refers to the traditional operating algorithm. Before conducting statistics using the histogram, there are several variants that simplify the feature types, each with their respective characteristics. These include rotational invariance (RI), uniform pattern 2 (U2), and a combination of rotational invariance with the uniform pattern 2 (RIU2).

The LBP-RI considers all rotated images within a 3×3 calculated window as having the same texture feature, resulting in statistics of shifting rotations, such as [00001111] and [1000111], being grouped together in a histogram. In contrast, the LBP-U2 treats rare textures in the image as the same feature type. The LBP-RIU2 combines the behaviours of both RI and U2. Although these three variants have distinct characteristics, we handle this aspect using a look-up table (LUT). This implies that the classic LBP, LBP-RI, LBP-U2, and LBP-RIU2 have the same operational complexity in terms of hardware design. The only variations lie in the width of the feature vector represented by the histogram and their classification accuracy. Specifically, the classic LBP, LBP-RI, LBP-U2, and LBP-RIU2 have feature vector widths of 256, 36, 59, and 10 in the histogram, respectively. A narrower feature vector width reduces the operational complexity in SVM.

### 3.3. Feature Vector Connected Method

Before training the SVM model with the texture feature vector, it is necessary to address an important issue regarding how these texture feature vectors should be connected. This is often overlooked in other studies, but it presents two or more different approaches. The first method tries to connect feature vectors from different OCT images, treating the model training as training with the entire OCT volume. This approach is illustrated in Figure 7a and referred to as ‘slice-chain’ method. The second method involves treating each OCT image as an independent dataset and predicting all OCT images individually. This approach is shown in Figure 7b and referred to as the ‘slice-threshold’ method. Ultimately, the final classification result is determined by comparing the model’s classification of whether the OCT image is diseased to a customised threshold within the OCT volume, as presented in [37].

After evaluating the hardware implementation, it was discovered that the operation matrix is almost the same size in both the slice-chain and slice-threshold methods, with the only difference being the column expansion or row expansion. Consequently, this indicates that the operational complexity is still significant.

Finally, we solved it with a different idea. Returning to the LBP statistic characteristic introduced in Section 3.2, we utilise the feature vector that serves as a counter for the occurrences of texture features in the image. We can extend the statistics of the texture feature to encompass the entire OCT volume, meaning that the histogram statistics are not limited to a single OCT image but instead scan the entire OCT volume. We can regard this as the sum of all occurrences of the same type of texture feature that are scattered across multiple OCT images within the OCT volume. The matrix, referred to as ‘slice-sum’, is shown in Figure 7c. This improved method significantly reduces the operational complexity by a factor of 100 compared to both slice-chain and slice-threshold methods. Moreover, its applicability may extend to other three-dimensional datasets with similar characteristics.

### 3.4. SVM Operation

Compared to the self-smart monitoring instrument, the automated AMD detection system is designed for mass-detection across various individuals, eliminating the need for online training. Moreover, this elimination of online training can significantly reduce area costs.

Based on this premise, the training of the SVM model was completed using the MATLAB toolbox, resulting in a fixed pre-trained model. The pretrained model parameters were then transmitted to our chip to predict the unknown OCT data.

With the exclusion of the SVM training step, the classifier can be obtained simply by using a linear SVM as follows:(3)$$y_{predict}=sign\left({Weight}_{j}\times {LBP\;Vector}_{j}\prime +Bias\right),$$
where ${LBP\;Vector}_{j}$ represents the feature vector derived from the histogram statistics of the LBP. ${Weight}_{j}$ is the weight vector, which is obtained as follows:(4)$${Weight}_{j}=\sum_{i=1}^{k}\left(\alpha_{i}\times y_{i}\times x_{i,j}\right).$$

Bias is the correction value, which is obtained as follows:(5)$$Bias=mean\left(y_{j}-x_{i,j}\times {Weight}_{j}\prime \right),$$
where k denotes the support vector point, $\alpha_{i}$ signifies the Lagrange multiplier, and $y_{i}$ represents the labels of the training matrix; $x_{i,j}$ signifies the samples of the training matrix obtained from the SVM toolbox as the model parameter. Additionally, $y_{predict}$ represents the final predicted result based on the positive or negative value of $\left({Weight}_{j}\times {LBP\;Vector}_{j}\prime +Bias\right)$. One represents normal OCT, and the other represents abnormal OCT, depending on the labelling during training.

The results of the kernel experiments are presented in Table 2. When comparing the different kernels to the Linear, Quadratic, and Cubic kernels, it is evident that they only led to a marginal accuracy improvement of less than 1% in OCT volume classification. Consequently, to simplify the hardware implementation, we prioritised the use of the Linear SVM.

### 3.5. System Simulation

To determine the final flow of the automated AMD detection system, we simulated three groups based on performance accuracy and hardware-friendly. The first group compared image size adjustments, the second group assessed noise removal techniques for the images, and the third group examined the effectiveness of the feature vector connected method and feature extraction using LBP, as shown in Table 3, Table 4 and Table 5.

Table 3 and Table 4 present four types of LBP for comparison, as reaching a firm conclusion regarding the performance accuracy of different LBP methods proved challenging. Among Group 1, LBP, LBP-RI, and LBP-RIU2 showed the highest accuracy in the non-adjusted image size flow. Although LBP-U2 had better accuracy in the crop flow, the 0.5% increase in accuracy was not worth implementing in the crop hardware.

Table 4 indicates that the accuracy in LBP, LBP-RI, and LBP-RIU2 is the highest when no noise removal is applied in the image flow. Only LBP-U2 exhibited better accuracy performance with the BM3DF. However, the use of BM3DF incurs significant hardware costs. Moreover, the accuracy of LBP-U2 with BM3DF is lower than that of LBP, LBP-RI, and LBP-RIU2 under the no-flow condition. Based on the findings from Table 4, the recommended approach is to utilise the image flow without any noise removal.

Table 5 presents the testing accuracy, which comprises 26 new datasets that were not used to train the model. Moreover, the testing detection process simulated the hardware behaviour described in the SVM detection model, as shown in Equation (3). Table 5 indicates that the slice-sum method displays better training accuracy compared to the slice-chain and slice-threshold methods for all LBP types. In addition, the testing accuracy of the slice-sum method was either superior or equal to that of the slice-chain and slice-threshold methods in LBP, LBP-RI, and LBP-U2. Therefore, we can conclude that the slice-sum method can replace the slice-chain and slice-threshold methods owing to its lower hardware complexity. Considering that LBP, LBP-RI, and LBP-U2 exhibit similar testing accuracy, we have chosen to implement the LBP-RI method owing to its narrower feature vector widths.

In conclusion, the final system flow does not involve image size adjustment or noise removal. Instead, we utilise the slice-sum method for feature connection and employ LBP-RI for feature extraction.

## 4. VLSI Implementation

This section provides a description of the hardware implementation in our chip, focusing on our objective of developing an automated eye disease system for embryos with an edged design. The input data consisted of pixel-by-pixel OCT images obtained directly from a portable OCT instrument, and the output was the predicted result.

### 4.1. System Architecture

An overview of the entire system architecture is depicted in Figure 8. Since the testing OCT image was processed pixel-by-pixel, a Line Buffer was employed to ensure that the LBP-RI accurately received the pixel information within a 3 × 3 calculated window cycle by cycle. The LBP Vector calculated the occurrence count of each type of feature vector and produced the final texture feature as an output. Statistic Saturation served to provide a warning effect. When the occurrence count of a specific feature vector type reached saturation, it was maintained, and the user was notified accordingly. It was crucial to carefully observe the detection results. Furthermore, the pre-trained parameter of the SVM model could be replaced by outside supply users to replace models in the future.

### 4.2. Line Buffer

The design that utilises a line buffer required a pre-processing time to allow for the reading of pixels from a specific location, facilitating the sequential transmission of an accurate 3 × 3 calculated window information to the LBP-RI cycle. The LBP-RI cycle was a long buffer arranged in series that stored all the scanned pixel values, as shown in the blue section of Figure 9. When the pixel read 2003, the line buffer could produce an output of [1 2 3 1001 1002 1003 2001 2002 2003], which was represented by the red border line. This output from the line buffer consisted of 72-bits. In the next cycle, the output would be [2 3 4 1002 1003 1004 2002 2003 2004]. However, it is important to note that this convenient operation comes with a significant area cost.

### 4.3. LBP-RI

The LBP-RI input from the line buffer consisted of nine pixels. These pixels included the neighbouring pixels, namely *g*_1_, *g*_2_, *g*_3_, *g*_4_, *g*_5_, *g*_6_, *g*_7_, and *g*_8_, excluding the centre pixel referred to as *gc*. To evaluate the texture feature, we employed an LUT that analysed the sign bit of the eight subtracted results. Figure 10 illustrates the register statistics that represent the occurrence frequencies of each feature type.

### 4.4. SVM Detection

The SVM detection performed a multiplication between the corresponding ${Weight}_{j}$ and ${LBPVector}_{j}$ and added the Bias in the final calculation. In our LBP-RI chip design, there were 36 weights and 36 feature types in the histogram statistics. To minimise the cost of multipliers, we used a single multiplier and summed every product over 36 cycles. Additionally, by positioning the Bias in the first addend of the adder, we successfully reduced the required number of cycles by one. The classification result was obtained from the sign bit of the sum at the end of the classifier, as shown in Figure 11.

### 4.5. Frequency Division Design

As the OCT volume represents a vast dataset, we utilised an example from Duke University for analysis. Each OCT volume consisted of 100 OCT images, and each image had sizes of 512 × 1000 pixels. This meant that a single OCT volume contained a total of 51,200,000 pixels that required processing. The clock cycle between the fast domain and the slow domain was 51,200,000:36. Based on this information, our objective was to improve the operating frequency during the pixel processing step and divide the frequency by six for the SVM detection step. This approach aimed to reduce the area cost, as depicted in Figure 12. Table 6 demonstrates that the frequency divider design decreases the area in the slow domain by 47.85%. Additionally, this design reduced power consumption by 79.71% under the same testing pattern.

### 4.6. Fixed-Point Simulation

To determine the appropriate word length for the hardware implementation, we needed to assess the classification accuracy of three components for simulating the fixed point: the LBPVector, Weight, and Bias. The analysis using floating-point numbers showed a classification accuracy of 92.31%. Our experiment consisted of a training dataset containing 100 AMD and 100 normal OCT volumes as well as a testing dataset with 13 AMD and 13 normal OCT volumes.

The LBPVector represented the histogram statistics derived from LBP-RI, which was a 1 × 36 integer vector. Both the Weight and Bias were parameters obtained from the pre-trained model in the MATLAB toolbox. Since the amount of integer data was too large, we truncated the fraction bits in both the Weight and Bias. Previous simulation results have confirmed this approach.

The process of conducting fixed-point operations started with the LBPVector, was followed by the Weight, and concluded with the Bias. However, when we reduced the integer bits to 23, the accuracy performance of the LBPVector significantly declined, as shown in Figure 13a. Furthermore, it was evident that the accuracy of all parameters dropped to 50%. In our simulation results using the testing dataset, all classification results predicted either ‘normal’ or ‘abnormal’ owing to the inability of the word length to express complete data information, as shown in Figure 13.

### 4.7. Implementation of ASIC

The chip implementation results using TSMC 40 nm are presented in Table 7. Both the line buffer and LBP-RI achieved a maximum operating frequency of 454.5 MHz. The SVM detection operated at 75.8 MHz, which had six times the frequency divider with a core area of 0.12 mm^2^. The line buffer represented the largest area cost, accounting for 88.8% of the overall core area. Regarding power consumption, the line buffer consumed 35.06 mW, the LBP consumed 1.65 mW, and the SVM consumed 0.03 mW. Figure 14 shows the layout view of the proposed detector.

### 4.8. Comparison

This section presents an overall comparison of various SVM classifier chips that utilised an ASIC implementation. To ensure a fair assessment, the definitions of the normalised evaluation equation are provided below, where S represents the technology and U represents the supply voltage.
(6)$${Normalised\;Area=Area\times \left(\frac{40}{S}\right)}^{2}$$
(7)$$Normalised\;Throughput=Throughput\times \frac{S}{40}\;$$
(8)$${Normalised\;Power=Power\times \left(\frac{0.9}{U}\right)}^{2}$$

Table 8 presents an overall comparison of the SVM classifier chip with other studies. The TCASII-2018 chip demonstrates extremely low power consumption and high throughput [29]. The TBCAS 2019 chip is unique, as it incorporates online training, making it a self-monitoring system that can obtain the same dataset from the same people. However, their work employed SMO, the training accelerator of the SVM model, which was their largest hardware design [5]. ICICDT 2022 achieves a maximum frequency of 500 MHz [38], whereas MWSCAS 2022 displays the highest accuracy, reaching 97.1% [39].

## 5. Conclusions and Future Work

In this study, an automated AMD detector has been proposed to assist ophthalmologists by reducing excessive and tedious medical examinations. Because the detector was expected to be implemented in a single edge-design chip and integrated into a portable OCT scanner, the LBP and SVM classifier were employed in the detector to minimise the hardware area cost. The slice-sum vector connection method was proposed to reduce the computational processing between the SVM classifier. Additionally, the frequency divider design was conducted to further reduce the area cost and computational power. Therefore, the area cost is reduced by 47.85% through optimisation of the slice-sum feature vector connection method, reduction of the multiplier and adder in the SVM detection block, and division of the clock domain to a slow frequency. Based on the proposed architecture, the area cost is only 0.12 mm^2^, making it easily applicable to edge chips. The power consumption is 39.43 mW, and the classification throughput reaches 8.87 decisions/s with a maximum frequency of 454.5 MHz. In terms of hardware acceleration, when using an OCT volume as the calculation standard, the hardware demonstrates a speed that is 139 times faster than the MATLAB software. This enables efficient handling of the OCT volume scanning speed by portable OCT scanners available in the market. Notably, the proposed detector is the first instance of single-chip implementation for automated AMD detection with a high classification accuracy of 92.31% in the literature. In the future, the proposed AMD detector design will be further fine-tuned, tested, and evaluated using a practical dataset extracted from a real portable OCT scanner, with the approval of an institutional review board. To further increase the classification accuracy, an improved automated AMD detector will be designed and implemented using other hardware-friendly ML or light-weight DL classifiers based on the fruitful single-edge chip design experiments demonstrated in this study.

## Figures and Tables

**Figure 1 sensors-23-07315-f001:**
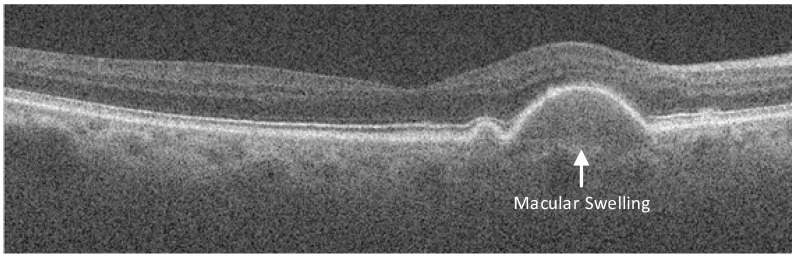
Macular swelling in an AMD patient, as observed through OCT.

**Figure 2 sensors-23-07315-f002:**
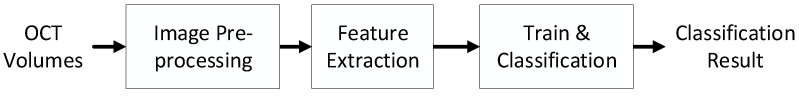
Basic flow of automated eye disease detection system.

**Figure 3 sensors-23-07315-f003:**
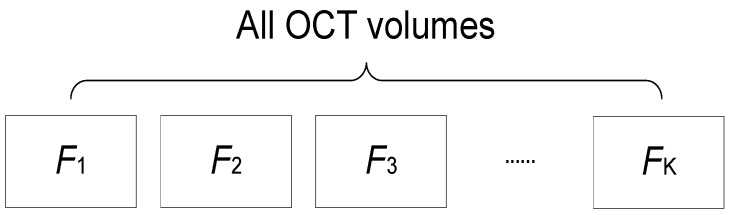
Schematic diagram of *K*-Fold cross-validation.

**Figure 4 sensors-23-07315-f004:**
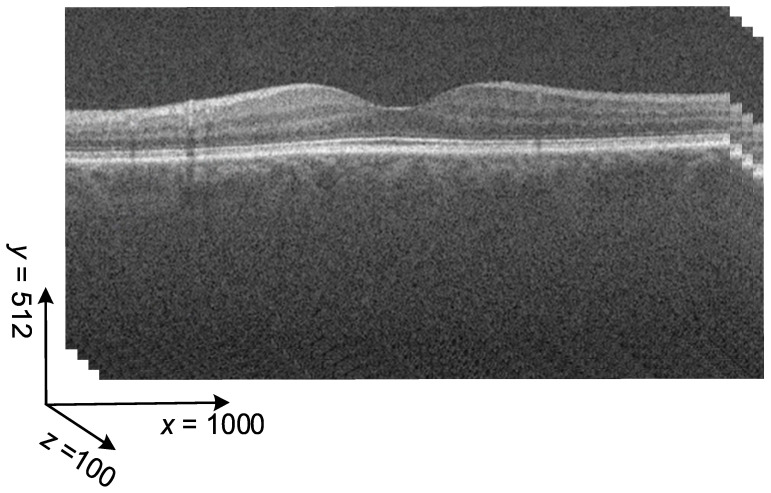
Concept of an OCT volume.

**Figure 5 sensors-23-07315-f005:**
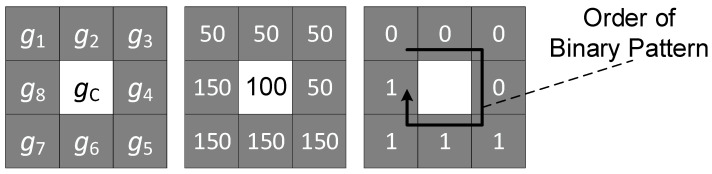
${LBP}_{8,1}$ derivation with a 3 × 3 calculated window.

**Figure 6 sensors-23-07315-f006:**
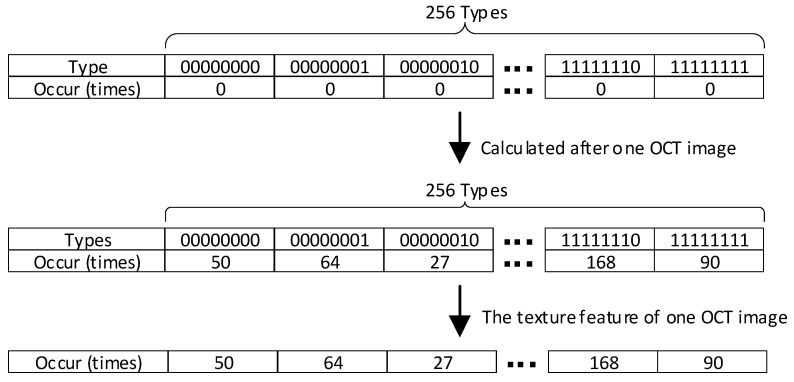
Texture feature of a single OCT image from the histogram statistics.

**Figure 7 sensors-23-07315-f007:**
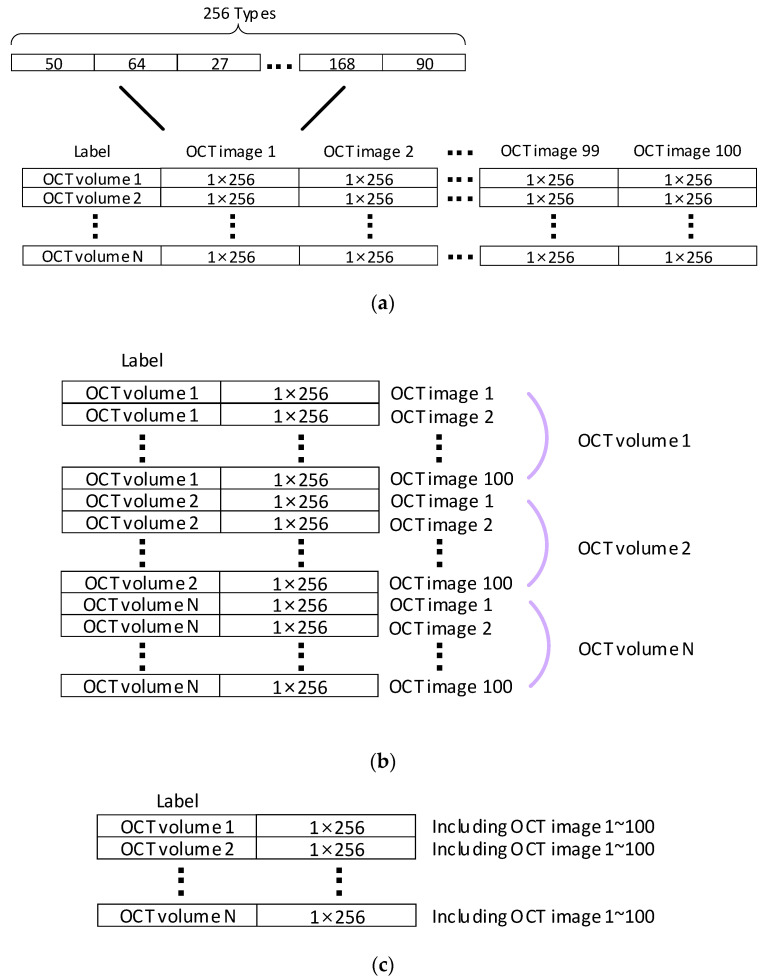
Overview of different connected methods: (**a**) slice-chain, (**b**) slice-threshold, and (**c**) slice-sum.

**Figure 8 sensors-23-07315-f008:**
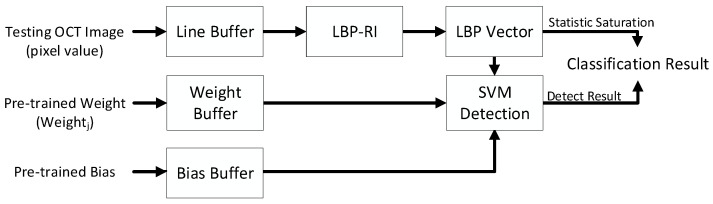
Overview of the proposed system architecture.

**Figure 9 sensors-23-07315-f009:**
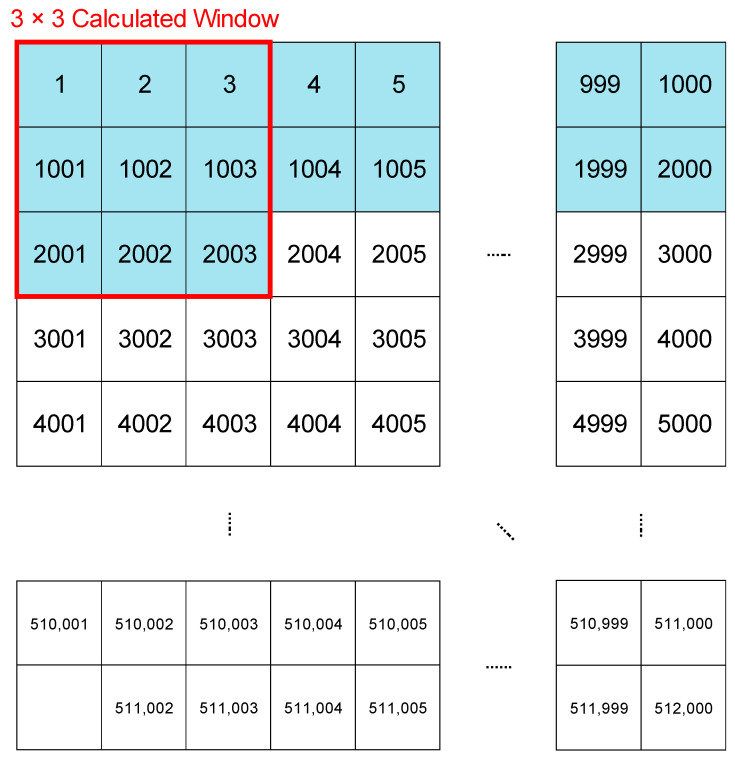
Line buffer exhibit in OCT image.

**Figure 10 sensors-23-07315-f010:**
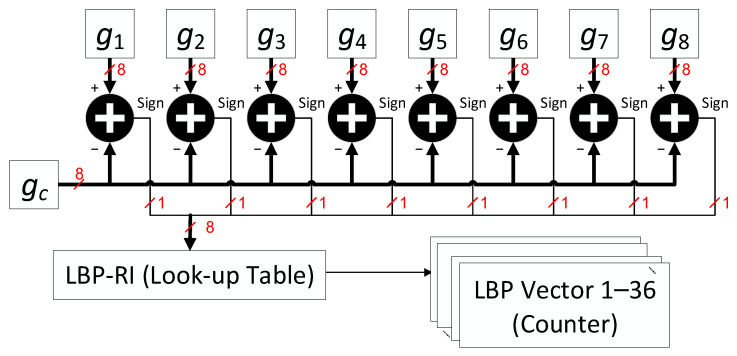
Hardware architecture of LBP-RI.

**Figure 11 sensors-23-07315-f011:**
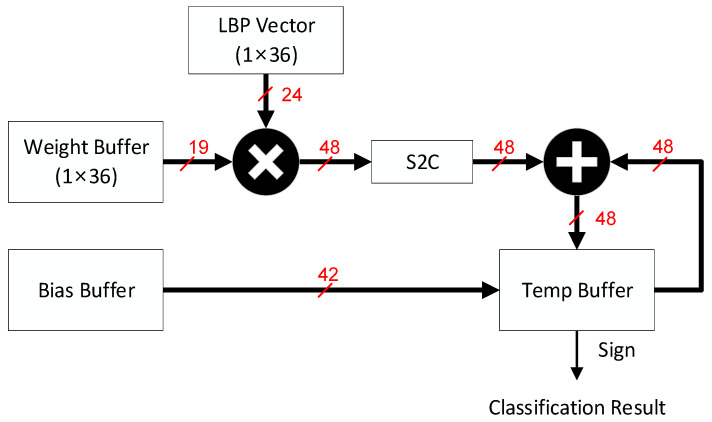
Hardware architecture of SVM Detection.

**Figure 12 sensors-23-07315-f012:**
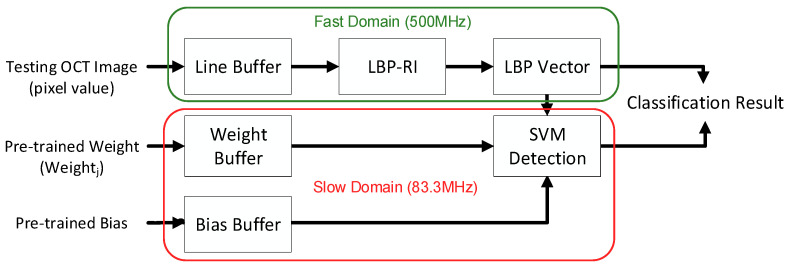
Overview of the detector with frequency divider.

**Figure 13 sensors-23-07315-f013:**
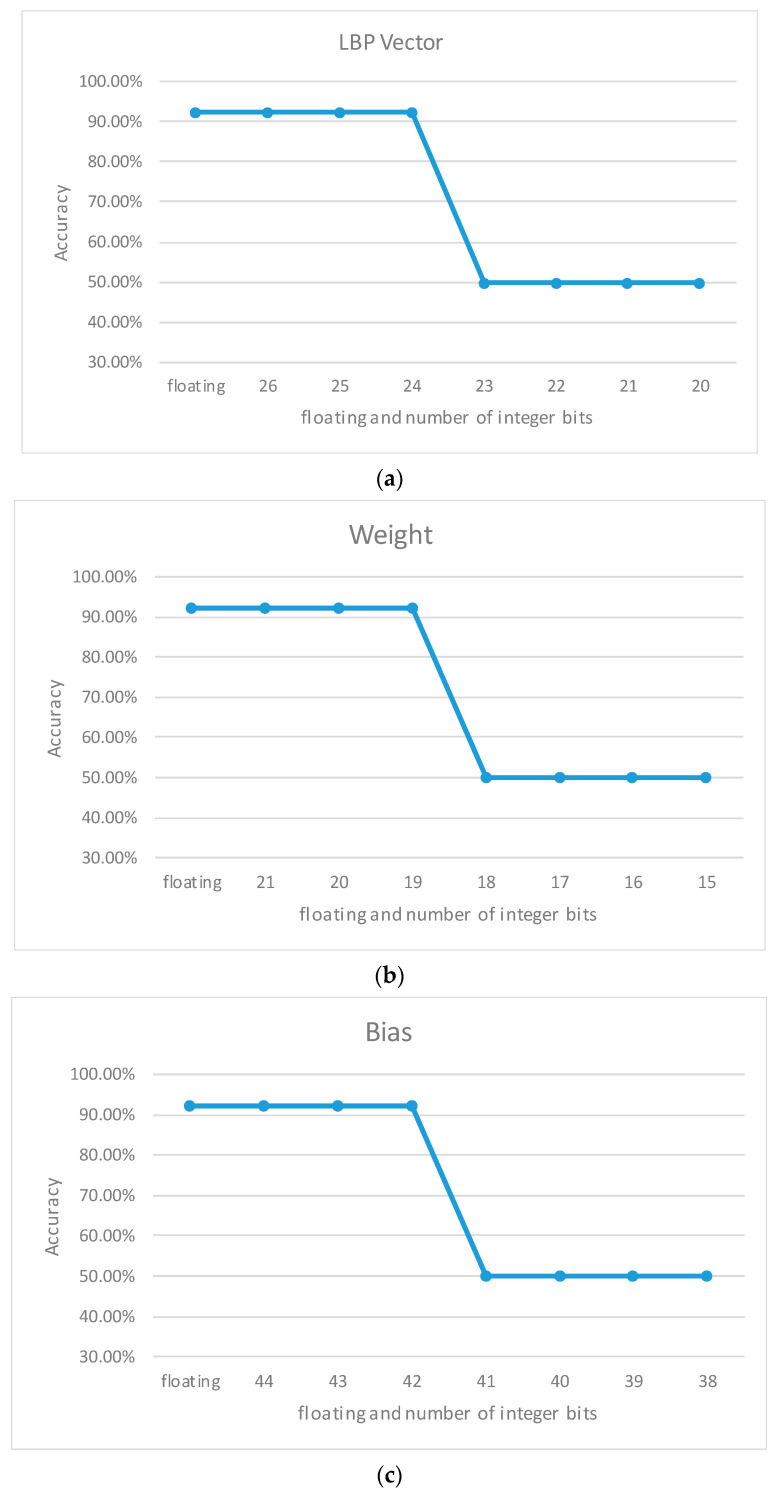
Overview of the fixed-point simulation (**a**) LBPVector, (**b**) Weight, and (**c**) Bias.

**Figure 14 sensors-23-07315-f014:**
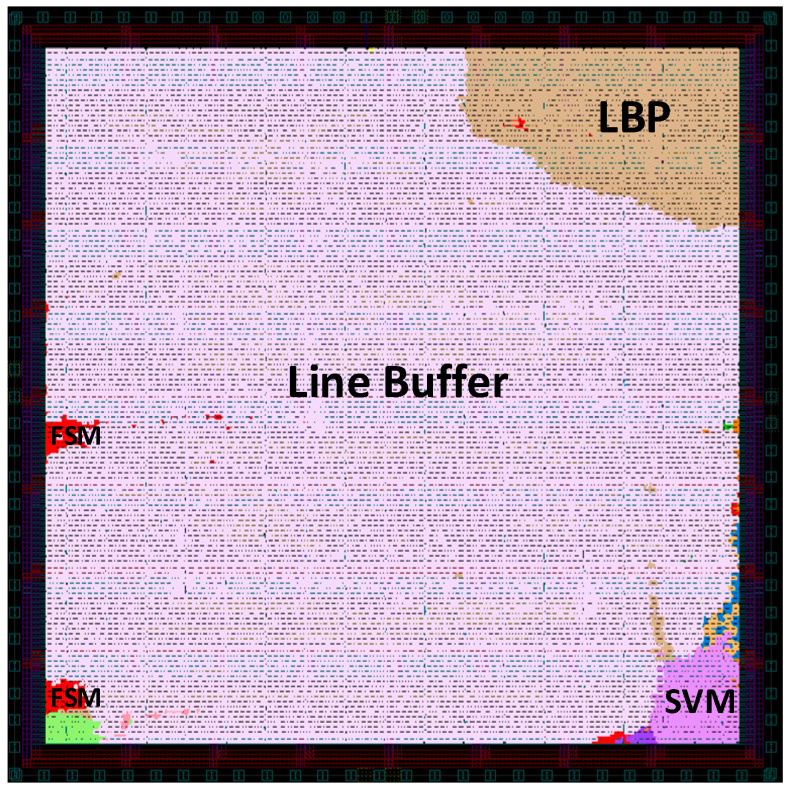
Chip layout of the proposed detector.

**Table 1 sensors-23-07315-t001:** Comparison of automated eye disease classification using SVM classifier.

Studies	Classification	Denoising	Feature Extraction	Performance
* 2011 [24]	Normal, AMD, ME, MH	Medium Filter	LBP	* AUC: 92.0%
2017 [25]	Normal, DME	BM3DF	LBP + HOG	SE: 87.5% SP: 87.5%
* 2017 [26]	Normal, AMD, DME	BM3DF	GoogLeNet	* ACC: 94.0%
2021 [27]	Normal, AMD	None	Proposed CNN	ACC: 97.7%
* 2023 [28]	Normal, DME, CNV, Drusen	None	DL-Based Feature Fution + HOG	* ACC: 99.9%
* 2023 [3]	AMD, DME, BRVO, CRVO, CSCR	None	DenseNet-201 + ACO	* ACC: 99.1%

* represents the image-level classification.

**Table 2 sensors-23-07315-t002:** Accuracy performance comparison in different SVM kernels using MATLAB Toolbox.

SVM Kernel	Linear	Quadratic	Cubic
Accuracy (%)	90.4	91.2	90.8

**Table 3 sensors-23-07315-t003:** Simulation results of Group 1 using MATLAB toolbox.

Flow (Group 1)		ACC (%)	SE (%)	SP (%)
Resize + LBP + SVM	LBP	87.70	86.84	88.60
	LBP-RI	85.50	86.84	84.21
	LBP-U2	83.30	82.46	84.21
	LBP-RIU2	89.00	87.72	90.35
Crop + LBP + SVM	LBP	86.40	88.60	84.21
	LBP-RI	87.30	87.72	86.84
	LBP-U2	85.10	80.70	89.47
	LBP-RIU2	86.80	88.60	85.09
None + LBP + SVM	LBP	89.00	92.98	85.09
	LBP-RI	87.70	90.35	85.09
	LBP-U2	84.60	83.30	85.96
	LBP-RIU2	90.80	93.85	87.72

**Table 4 sensors-23-07315-t004:** Simulation results of Group 2 using MATLAB toolbox.

Flow (Group 2)		ACC (%)	SE (%)	SP (%)
Resize + LBP + SVM	LBP	86.40	89.47	83.33
	LBP-RI	85.50	88.60	82.46
	LBP-U2	87.30	91.23	83.33
	LBP-RIU2	88.60	92.11	85.09
Crop + LBP + SVM	LBP	87.70	90.35	85.09
	LBP-RI	83.30	82.46	84.21
	LBP-U2	86.80	89.47	84.21
	LBP-RIU2	87.70	89.47	85.96
None + LBP + SVM	LBP	89.00	92.98	85.09
	LBP-RI	87.70	90.35	85.09
	LBP-U2	84.60	83.30	85.96
	LBP-RIU2	90.80	93.85	87.72

**Table 5 sensors-23-07315-t005:** Simulation results of Group 3 using MATLAB toolbox.

Flow (Group 3)		Training Accuracy (MATLAB Toolbox) (%)	Testing Accuracy (Hardware-Based Linear-SVM) (%)
Slice-chain	LBP	87.50	92.31
	LBP-RI	83.50	88.46
	LBP-U2	88.50	92.31
	LBP-RIU2	83.00	76.92
Slice-threshold	LBP	84.20	92.31
	LBP-RI	77.90	88.46
	LBP-U2	82.00	92.31
	LBP-RIU2	74.40	84.62
Slice-sum	LBP	89.50	92.31
	LBP-RI	87.00	92.31
	LBP-U2	89.00	92.31
	LBP-RIU2	84.50	76.92

**Table 6 sensors-23-07315-t006:** Comparison of AMD detector with frequency divide version.

		Original AMD Detector	AMD Detector with Frequency Divider

Frequency (MHz)		500.0	500.0/83.3
Latency (ns)		102,400,072	102,400,434
Total Power of Slow Domain (mW)		0.25783 (100%)	0.05232 (20.29%)
Total Area of Slow Domain (μm^2^)		6251.74 (100%)	3260.25 (52.15%)

**Table 7 sensors-23-07315-t007:** Summary of the chip used for the proposed detector.

Technology	TSMC 40 nm CMOS		
Logic Utilisation	89.33%		
Supply Voltage (V)	0.9		
Core Area (mm^2^)	0.12 (0.35 × 0.35)		
Logic Gate Count	177 k		
Block	Line Buffer	LBP-RI	SVM Detection
Max. Frequency (MHz)	454.5	454.5	75.8
Area (mm^2^)	0.09 (88.8%)	0.0088 (8.2%)	0.0017 (0.2%)
Power (mW)	35.06	1.65	0.03

**Table 8 sensors-23-07315-t008:** Overall comparison of the SVM classifier against other chips.

	This Work	TCASII’18 [29]	TBCAS’19 [5]	ICICDT’22 [38]	MWSCA’22 [39]
Technology	TSMC 40 nm	TSMC 65 nm	UMC 65 nm	TSMC 65 nm	TSMC 40 nm
Object	AMD Detection	Personalised Stress Detection	Neural Seizure Detection	Human Detection	Arrhythmia Detection
Online Training	No	No	SMO	No	No
Core Area (mm^2^)	0.12	0.17	0.20	1.14	0.54
Max Frequency (MHz)	454.5	250	100	500	100
Power (mW)	39.43	7.6 × 10^−7^	14.91	195	5.7 × 10^−6^
Accuracy (%)	92.31	96.7	90.34	N/A	97.1
Frame Rate (fps)	887	N/A	N/A	139	N/A
Throughput (decision/s)	8.87	243,902	N/A	N/A	N/A
Normalised Area (mm^2^)	0.12	0.06	0.08	0.43	0.54
Normalised Frame Rate (fps)	887	N/A	N/A	255.88	N/A
Normalised Throughput (decision/s)	8.87	396340.75	N/A	N/A	N/A
Normalised Power (mW)	39.43	6.2 × 10^−7^	12.08	157.95	5.7 × 10^−6^

Frame Rate = image/s; Throughput = decision/s. (One decision includes 100 OCT images.). N/A: Not Available.

## Data Availability

Publicly available datasets were analyzed in this study. This data can be found here: [https://people.duke.edu/~sf59/software.html].

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
