# Peer review of "Automated Age-Related Macular Degeneration Detector on Optical Coherence Tomography Images Using Slice-Sum Local Binary Patterns and Support Vector Machine"

_sensors, 2023, doi:10.3390/s23177315_

Round 1

Reviewer 1 Report

Dear Authors,

This work contains essential material, useful when solving practical and possibly future theoretical problems in application of artificial intelligence, namely local binary patterns and support vector machines. Particular attention is paid to methods for reducing the computational complexity of  SVM-based detection. The authors analyzed feature extraction using existing methods and proposed VLSI architecture to implement their approach. The description of the chip in the automatic system for detection of desease of the retina of the eye that occurs with aging is given.  The data used in the work are obtained by optical coherence tomography and a new economical slice sum scheme for data organization is proposed. The rationale for this scheme is given on the basis of the statistical properties of histograms of indivdual layers, chain of layers, and OCT volume in general. As a result, greater classification accuracy has been achieved. The article is well illustrated, which helps the reader. In my opinion, the work deserves publication.

Author Response

Responses to Reviewer #1

Thanks for your time and efforts evaluating our paper.

  1. Reviewer #1 states, “This work contains essential material, useful when solving practical and possibly future theoretical problems in application of artificial intelligence, namely local binary patterns and support vector machines. Particular attention is paid to methods for reducing the computational complexity of SVM-based detection. The authors analyzed feature extraction using existing methods and proposed VLSI architecture to implement their approach. The description of the chip in the automatic system for detection of desease of the retina of the eye that occurs with aging is given. The data used in the work are obtained by optical coherence tomography and a new economical slice sum scheme for data organization is proposed. The rationale for this scheme is given on the basis of the statistical properties of histograms of indivdual layers, chain of layers, and OCT volume in general. As a result, greater classification accuracy has been achieved. The article is well illustrated, which helps the reader. In my opinion, the work deserves publication.”

Thank you for pointing out the strength of this paper. We appreciate your favorable review and efforts evaluating our paper.

Reviewer 2 Report

Thanks for the detailed paper, I have read it. I suggest that the author should revise the manuscript before resubmission. My main concern and comments are listed as follows:

1. The Literature Review section is missing. The author should present a critical review of various state-of-the-art methods. Also, it would be beneficial to recognize the gaps, pros, and cons of the corresponding techniques to better present the necessity of the work.

2. Currently, deep learning techniques are widely employed in many fields. Please compare your method with some outstanding deep-learning such as VggNet, Resnet, and GoogLeNet.

3. What are the limitations of the proposed method that motivated the current research? It should be mentioned in the paper.

4. What is the future scope of the current work?

5. Hyperspectral and multispectral systems also play an important role in disease diagnosis. For example, Smartphone imaging spectrometer for egg/meat freshness monitoringand Open-source mobile multispectral imaging system and its applications in biological sample sensing, it is suggested that hyperspectral and multispectral systems should be discussed.

6. The authors should discuss more recent methods which were published in refereed international journal articles in the recent three years.

Author Response

Responses to Reviewer #2

Thanks for your elaborate review and the favorable comments. Your comments are very helpful in improving the quality of this submission. We have revised this submission based on your comments. We address your technical concerns as follows. (To address the following comments, the cited sections, figures, and reference numbers are in the revised paper.)

  1. Reviewer #2 states, “The Literature Review section is missing. The author should present a critical review of various state-of-the-art methods. Also, it would be beneficial to recognize the gaps, pros, and cons of the corresponding techniques to better present the necessity of the work.”

Thank you for pointing the weakness of this paper out. The review section has been added in Section 2.1 (Related Works). Please read it for details. We hope this section can adequately address your concern.

  1. Reviewer #2 states, “Currently, deep learning techniques are widely employed in many fields. Please compare your method with some outstanding deep-learning such as VggNet, Resnet, and GoogLeNet.”

In our previous publication [6], we compared the SVM classifier with some outstanding deep-learning (DL) classifiers, such as AlexNet, GoogLeNet, and Inception-ResNet-v2. The AlexNet, GoogLeNet, and Inception-ResNet-v2 achieved high AMD detection accuracies up to 97.39%, 96.41%, and 98.18%, respectively. However, the study in [6] also revealed that the DL-based disease classification consumes high computational complexity, using powerful graphics processing units (GPUs) of NVDIA GeForce GTX 1070, to achieve the high detection accuracy compared to the SVM-based disease classification using the Intel i7-8700 3.2GHz 6-core processor only. It means that the current DL-based AMD detections executed by high-level GPUs are not cost-effective hardware implementation. This initially motivated us to design a low-cost automated AMD detector using the SVM classifier since the SVM exhibits a lower area cost and becomes suitable for single edge-chip implementation [5], [29]. Additionally, the study in [6] can be regarded as our prior research before this submission. As mentioned in the response to your 3rd comment, we focus on developing a low-cost single-chip automated AMD detector in this submission. The outstanding DL techniques are not cost-effective but computation-intensive. Compared with the computation-intensive DL techniques, the proposed architecture has been successfully implemented in a 0.12-mm2 single chip (see Fig. 14 and Table 7) and achieved a classification accuracy of 92.31%. The classification throughput reaches 8.87 decisions/sec at a maximum frequency of 454.5 MHz. To the best of our knowledge, this submission presents the first instance of single-chip implementation (see Fig. 14 and Table 7) for automated AMD detection in the literature. In terms of hardware acceleration, when using an OCT volume as the calculation standard, the hardware demonstrates a speed that is 139 times faster than the MATLAB software.   

Thank you for pointing this out. Based on your 1st and this comments, we have added some statements about DL-based disease classification in Section 2.1. We hope the above modifications can adequately address your concern.

  1. Reviewer #2 states, “What are the limitations of the proposed method that motivated the current research? It should be mentioned in the paper.”

Thank you for pointing the weakness of this paper out. Based on this comment, the 4th - 5th paragraphs in Section I (Introduction) have been significantly rewritten to describe the limitations of the proposed method that motivated the current research and highlight the major contributions of the proposed AMD detector. As mentioned in the response to your 2nd comment, the current DL-based AMD detections executed by high-level GPUs are not cost-effective hardware implementation. This initially motivated us to design a low-cost automated AMD detector using the SVM classifier since the SVM exhibits a lower area cost and becomes suitable for single edge-chip implementation [5], [29]. In this submission, we focus on developing an automated AMD detector that can be implemented in a low-cost edge-design chip and integrated with a portable OCT scanner for supporting medical service in rural areas. However, the current DL-based AMD detections (AlexNet, GoogLeNet, and Inception-ResNet-v2) executed by a computation-intensive GPU are not cost-effective Implementations for the edge-design chip. (Note that this can be also revealed in our publication [6]). Therefore, the proposed automated AMD detector utilizes the proposed slice-sum LBP and SVM classifier to achieve low-complexity and hardware-friendly hardware architectures.

We hope the modifications can adequately address your concern.

  1. Reviewer #2 states, “What is the future scope of the current work?

Thank you for pointing the weakness of this paper out. The future scope of this work is listed as follows:

“In the future, the proposed AMD detector design will be further fine-tuned, tested, and evaluated using a practical dataset extracted from a real portable OCT scanner, with the approval of an institutional review board.”

“To further increase the classification accuracy, an improved automated AMD detector will be designed and implemented using some other hardware-friendly ML or light-weight DL classifiers based on the fruitful single-edge-chip design experiments demonstrated in this study.”

Based on this comment, Section 5 (Conclusions and Future Work) has been rewritten. Please read it for details. We hope the modifications can adequately address your concern.

  1. Reviewer #2 states, “Hyperspectral and multispectral systems also play an important role in disease diagnosis. For example, “Smartphone imaging spectrometer for egg/meat freshness monitoring” and “Open-source mobile multispectral imaging system and its applications in biological sample sensing”, it is suggested that hyperspectral and multispectral systems should be discussed.”

Thanks for pointing this out. We have added “Open-source mobile multispectral imaging system and its applications in biological sample sensing” in Reference [11] in this submission. The related statement has also been added to the 2nd paragraph of Section I (Introduction). “Smartphone imaging spectrometer for egg/meat freshness monitoring” is not cited in this submission because we think the application of egg/meat freshness monitoring would not be related to this submission.

  1. Reviewer #2 states, “The authors should discuss more recent methods which were published in refereed international journal articles in the recent three years”

Thank you for pointing the weakness of this paper out. Based on your 1st and this comments, several studies have been added to Reference and discussed in the submission. References [1]-[3], [6], [8], [11], [17], [19]-[22], [27], [28] are refereed international journal articles in 2020 - 2023. References [3], [6], [8], [17], [19]-[22], [27], [28] have been majorly discussed in Section 2.1 (Related Works). We hope the modifications can adequately address your concern.

Reviewer 3 Report

Dear Authors,

   Thank you for your manuscript submission to MDPI J. Sensors. This paper proposed slice-sum, a modified feature vector connected method to reduce computational complexity for OCT images. The AI-based topic on age-related macular degeneration (AMD) detection system is also intriguing. After overall inspection, this paper is basically complete and displayed relatively good set of work, it is clearly organized and well written. Suggested aspects for further edits are presented as follows:

   a) Abstract and Keywords: Very good presentation. In Line 31, consider adding "slice-sum" as the fifth keyword. 

   b) Introduction: Supplementing the cited reference (if any) in the first 4 paragraphs. In Lines 77-94, consider rewritting this part into a brief summary on the major contributions of the proposed AMD detection system, in 3-4 manifolds. Lines 99-100, the sentence for final section needs revision

   c) This paper is absent from a section of related work, despite some preservations, the authors may consider if it can be inserted above Line 101.

   d) Lines 140-146: Please italize each "K" of K-fold and notations for "F" of Fi, and horizontally stretch Fig. 1 to fix the minor distortions. Thanks a lot!

   e) Section 3: Lines 155-156, and Lines 171-172, I think proper size and middle alignments need to be applied to Figs. 3-4, respectively.

   f) In Lines 273-276, Lines 290-291, each of the numeric scores in Tables 2-4, are better to be uniformly rounded with 2-valid digits after each decimal. (i.e., 89-->89.00)

   g) Other figures and tables: Please be sure each one is arranged with the proper font style, size and locations, image resolution needs to be enhanced, check Table 5 in Lines 355-356 and see if any trivial issues co-exist.

   h) Line 376, replace "ASIC Implement" with "Implementation of ASIC". Line 502, replace "Conclusions" with "Conclusions and Future Work".

   i) Section 5: It is better to present the last section in 3 paragraphs.  The authors are expected to be more specific when summarize the entire set of work. After comprising a main summary of accomplished tasks, the keynote quantitative results and crucial statements in concluding remarks; then the potential areas of challenging issues need to be proposed with discussions according to a variety of suggested open topics. Lastly, you may present some specific research plans to solve the proposed topics when prospecting, or the prospective orentations in future investigation. 

   j) References: Some edits must be proceeded regarding current version. i) Abbreviations on title of some conference proceedings, i.e., [5], [9], [17], [20], and [21]. ii) Citing a few more classical OCT-based image analysis schemes in the past decade as well as a few highly visible medical journal articles. iii) Enhance the cited reference on the state-of-the-art approaches on machine learning, deep learning, and AI-based reinforce learning and aid-assisted decision making schemes on clinical OCT image analysis.

   Some other potential minor defects must be fixed in the revised version: 

   1)There are still a few awkward narrations in this manuscript, please polish the content in some subsections, and proceed with expected grammatical edits and linguistic calibrations when fixing all the other related problems.

   2) Stop minor hypherating issues when crossing over two adjacent lines. Consider editing an MS word version of MDPI template to resolve the issue.

   Best of luck for your revisions.  Please update at your best convenience. Thanks a lot! 

With warm regards,

Yours sincerely,

Quality of English writing of this paper is acceptable. There are still some room for further improvement. 

Author Response

Responses to Reviewer #3

Thanks for your elaborate review and the favorable comments. Your comments are very helpful in improving the quality of this submission. We have revised this submission based on your comments. We address your technical concerns as follows. (To address the following comments, the cited sections, figures, and reference numbers are in the revised paper.)

  1. Reviewer #3 states, “Thank you for your manuscript submission to MDPI J. Sensors. This paper proposed slice-sum, a modified feature vector connected method to reduce computational complexity for OCT images. The AI-based topic on age-related macular degeneration (AMD) detection system is also intriguing. After overall inspection, this paper is basically complete and displayed relatively good set of work, it is clearly organized and well written.”

Thank you for pointing out the strength of this paper.  

  1. Reviewer #3 states, “Suggested aspects for further edits are presented as follows: Abstract and Keywords: Very good presentation. In Line 31, consider adding "slice-sum" as the fifth keyword.”

Thank you for pointing this out. “Slice-sum” has been added to the fifth keyword.

  1. Reviewer #3 states, “Introduction: Supplementing the cited reference (if any) in the first 4 paragraphs. In Lines 77-94, consider rewritting this part into a brief summary on the major contributions of the proposed AMD detection system, in 3-4 manifolds. Lines 99-100, the sentence for final section needs revision”

Thank you for pointing the weakness of this paper out. Based on this comment, eleven studies have been added to the reference and cited in this submission to support the statements in Section 1. Section 1 (Introduction) has been significantly rewritten to highlight the major contributions of the proposed AMD detection system. The last sentence of Section 1 for the final section has been revised. Please read Section 1 for details.

  1. Reviewer #3 states, “This paper is absent from a section of related work, despite some preservations, the authors may consider if it can be inserted above Line 101.”

Thank you for pointing the weakness of this paper out. The section of related work has been added in Section 2.1 (Related Works). Please read it for details. We hope this section can adequately address your concern.

  1. Reviewer #3 states, “Lines 140-146: Please italize each "K" of K-fold and notations for "F" of Fi, and horizontally stretch Fig. 1 to fix the minor distortions. Thanks a lot!”

Thanks for pointing this out. They have been italicized. Fig. 1 has been horizontally stretched to restore its original aspect ratio.

  1. Reviewer #3 states, “Section 3: Lines 155-156, and Lines 171-172, I think proper size and middle alignments need to be applied to Figs. 3-4, respectively.”

Thanks for pointing this out. They have been adjusted.

  1. Reviewer #3 states, “In Lines 273-276, Lines 290-291, each of the numeric scores in Tables 2-4, are better to be uniformly rounded with 2-valid digits after each decimal. (i.e., 89-->89.00)”

Thanks for pointing this out. Tables 2-4 have been modified.

  1. Reviewer #3 states, “Other figures and tables: Please be sure each one is arranged with the proper font style, size and locations, image resolution needs to be enhanced, check Table 5 in Lines 355-356 and see if any trivial issues co-exist.”

Thanks for pointing the weakness of this paper out. All figures and tables have been checked to make sure each one is arranged with the proper font style, size, locations, and image resolution. Table 5 has been revised to remove the trivial data.

  1. Reviewer #3 states, “Line 376, replace "ASIC Implement" with "Implementation of ASIC". Line 502, replace "Conclusions" with "Conclusions and Future Work".”

Thanks for pointing this out. “ASIC Implement” and "Conclusions" have been replaced by “Implementation of ASIC” with "Conclusions and Future Work", respectively.

  1. Reviewer #3 states, “Section 5: It is better to present the last section in 3 paragraphs. The authors are expected to be more specific when summarize the entire set of work. After comprising a main summary of accomplished tasks, the keynote quantitative results and crucial statements in concluding remarks; then the potential areas of challenging issues need to be proposed with discussions according to a variety of suggested open topics. Lastly, you may present some specific research plans to solve the proposed topics when prospecting, or the prospective orentations in future investigation.”

Thanks for pointing the weakness of Section 5 out. Following the same style in [2] and [3] that are published in the same journal, MDPI sensor, Section 5 still contains one paragraphs. Based on your constructive suggestion, however, we present Section 5 in 3 parts listed as follows.

  • (1st Part: summary of accomplished tasks) In this study, an automated AMD detector has been proposed to assist ophthalmologists by reducing the excessive and tedious medical examinations. Because the detector was expected to be implemented in a single edge-design chip and integrated into a portable OCT scanner, the LBP and SVM classifier were employed in the detector to minimise the hardware area cost. The slice-sum vector connection method was proposed to reduce the computational processing between the SVM classifier. Additionally, the frequency divider design was conducted to further reduce the area cost and computational power.
  • (2nd Part: keynote quantitative results) Therefore, the area cost is reduced by 47.85% through optimisation of the slice-sum feature vector connection method, reduction of the multiplier and adder in the SVM detection block, and division of the clock domain to a slow frequency. In terms of hardware acceleration, when using an OCT volume as the calculation standard, the hardware demonstrates a speed that is 139 times faster than the MATLAB software. Based on the proposed architecture system, the area cost is only 0.12 mm2, making it easily applicable to edge chips. The power consumption is 39.43 mW, and the classification throughput reaches 8.87 decisions/sec with a maximum frequency of 454.5 MHz. This enables efficient handling of the OCT volume scanning speed by portable OCT scanners available in the market. Notably, the proposed detector is the first instance of single-chip implementation for automated AMD detection with a high classification accuracy of 92.31% in the literature.
  • (3rd Part: prospective orientations in future investigation) In the future, the proposed AMD detector design will be further fine-tuned, tested, and evaluated using a practical dataset extracted from a real portable OCT scanner, with the approval of an institutional review board. To further increase the classification accuracy, an improved automated AMD detector will be designed and implemented using other hardware-friendly ML or light-wight DL classifiers based on the fruitful single-edge-chip design experiments demonstrated in this study.

The above statements have been written in Section 5. Please read it for details. We hope the modifications can adequately address your concern.

  1. Reviewer #3 states, “References: Some edits must be proceeded regarding current version. i) Abbreviations on title of some conference proceedings, i.e., [5], [9], [17], [20], and [21]. ii) Citing a few more classical OCT-based image analysis schemes in the past decade as well as a few highly visible medical journal articles. iii) Enhance the cited reference on the state-of-the-art approaches on machine learning, deep learning, and AI-based reinforce learning and aid-assisted decision making schemes on clinical OCT image analysis.”

Thanks for pointing this out. Abbreviations on title of some conference proceedings have been added. Based on your 3rd, 4th, and this comments, several studies have been added to Reference and discussed in the submission. Several classical OCT-based image analysis schemes in the past decade as well as a few highly visible medical journal articles [3], [6]-[8], [16]-[28] are cited in the submission. The state-of-the-art approaches on the AI-based decision-making schemes on clinical OCT image analysis have been majorly discussed in Section 2.1 (Related Works). We hope the modifications can adequately address your concern.

  1. Reviewer #3 states, “Some other potential minor defects must be fixed in the revised version: There are still a few awkward narrations in this manuscript, please polish the content in some subsections, and proceed with expected grammatical edits and linguistic calibrations when fixing all the other related problems.”

Thanks for pointing this out. To improve the language clarity of the whole manuscript, an editor from the Editage, a professional editing service company, has helped to proofread this manuscript. The editor is a native speaker of English. In addition, we have also proofread this manuscript again after the editing made by the editor.

  1. Reviewer #3 states, “Stop minor hypherating issues when crossing over two adjacent lines. Consider editing an MS word version of MDPI template to resolve the issue.”

Thanks for pointing this out. This issue has been resolved.

  1. Reviewer #3 states, “Comments on the Quality of English Language Quality of English writing of this paper is acceptable. There are still some room for further improvement.”

Thanks for pointing this out. We have addressed this in the response to your 12th comment. Please it for details.

Round 2

Reviewer 2 Report

Accept in present form